# Motivation Aware Question Decomposition: An approach to Debugging Reasoning in Vision-Language Models

## Abstract

Vision-Language Models (VLMs) have achieved impressive results in Visual Question Answering (VQA), yet they remain prone to hallucination – generating plausible but visually unsupported answers. Existing approaches have attempted to mitigate hallucination by introducing multi-turn VQA, where the model answers intermediate sub-questions or follows step-by-step reasoning. In this work, we explore a fundamental and underexamined question: where should a model look during multi-turn VQA when a visual inventory is available? We propose Reflection with Visual Inventory (RVI), a cognitively inspired framework that structures visual reasoning through iterative question decomposition and localized image inspection. Rather than treating the image as a single static input, our RVI builds and maintains a Visual Inventory—a dynamic collection of semantically relevant image crops that direct attention and support answer verification throughout the reasoning process. At each step, the system poses binary sufficiency queries to determine whether the current sub-question can be resolved using the existing inventory. If sufficiency fails, the model reflects by updating the inventory, emulating human-like visual reasoning and self-correction. RVI builds visual grounding into each step of reasoning, moving beyond static or post-hoc grounding and giving clear, step-by-step supervision. RVI makes errors like poor decomposition or weak grounding visible, giving clear signals that help systematically debug VLM reasoning. We demonstrate that integrating RVI into multiple VLM architectures improves performance on VQA instances from GQA and A-OKVQA datasets where baseline models fail, highlighting its effectiveness in reducing hallucinations and enhancing answer fidelity.

## 1 Introduction

Vision-Language Models (VLMs) have made significant progress on tasks such as Visual Question Answering (VQA), image captioning, and multimodal dialogue. Despite these gains, they remain susceptible to hallucination—producing confident, plausible answers that are unsupported by the visual input. Such behavior poses challenges in safety-critical domains like medical imaging, robotics, and autonomous systems. Recent work attributes hallucination to over-reliance on language priors Lin et al. (2024); Li et al. (2023b), weak visual grounding Geigle et al. (2024); Phukan et al. (2025), and the absence of structured reasoning Yao et al. (2024); Tascon-Morales et al. (2023).

Efforts to reduce hallucination include grounding supervision Lee et al. (2024); An et al. (2025), post-hoc verification Otani et al. (2023), and self-refinement through self-consistency or feedback Wang et al. (2023); Madaan et al. (2024). However, these methods generally operate as post-hoc corrections: they adjust or filter predictions after reasoning has already occurred, rather than guiding the reasoning process itself. As a result, they may suppress visible errors but leave the underlying failure modes intact.

Existing decomposition methods often treat breakdown as a syntactic exercise, overlooking the motivation behind the question. Yet questions vary in intent: some seek identity, others description, purpose, or interest, each requiring different evidence Naik et al. (2023); Khan et al. (2023); Zhang et al. (2024). When motivation is ignored, sub-questions become brittle or misleading. For example,

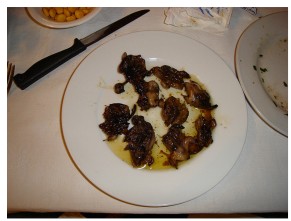

| Question: "Which food item is the knife used for?" | Inferred Motivation: *Identify the specific food on the table that the knife is intended to cut or serve.* |
|---|---|
| **Motivation-unaware sub-questions** 1. Where is the knife located? 2. How is the knife being used? 3. Which food item is the knife used for? | **Motivation-aware sub-questions** 1. What type of knife is this? 2. What is the primary use of this knife? 3. In what cuisine might this knife commonly used? 4. What food items are typically cut with this knife? |

Figure 1: Motivation-awareness leads to principled decomposition of the question into relevant sub-questions that help answer the original question. Both subquestions and motivation are generated using Qwen2.5-VL-7B-Instruct Team (2024).

as shown in Figure 1, for the query "Which food item is the knife for?", model generates irrelevant attribute questions about location of the knife and how the knife is being used when it is lying on the side of the plate. Such errors expose a deeper problem: when asked about motivation or intention, VLMs often lean on learned language priors and sometimes answer as though no image were provided at all Lin et al. (2024); Li et al. (2023b). This reflects biases inherited from datasets that recycle question patterns across benchmarks Geigle et al. (2024); Phukan et al. (2025). As a result, failure points can arise at multiple levels: misinterpreting what the question is asking (motivation), taking a shortcut with priors, decomposing into irrelevant sub-questions, failing to ground visually, or ignoring evidence already gathered Yao et al. (2024); Tascon-Morales et al. (2023). In contrast, motivation-aware decomposition aligns sub-questions with the reason the query is asked, ensuring that evidence is semantically relevant and making the reasoning both interpretable and debuggable.

To design such principled decomposition, we draw on cognitive science, which has long studied how humans break down complex tasks, manage limited attention, and align reasoning with goals. Three foundational principles guide our approach. First, *focused attention* enables humans to selectively process task-relevant visual information Cohen & Cohen (2014). Second, *dual coding theory* posits that cognition integrates verbal and visual pathways for more robust reasoning Paivio (1971; 1990). Third, *working memory* constraints motivate decomposing complex tasks into manageable sub-tasks Baddeley (2020). To these, we add a fourth principle: *motivation awareness*, the idea that sub-questions should reflect not only visual detail but also the underlying intent of the main query, thereby aligning decomposition with the reasoning goal.

We propose Reflection with Visual Inventory (RVI), a cognitively inspired framework for motivation-aware, multi-step reasoning in VQA. RVI begins by explicitly inferring the motivation behind a question—why it is being asked and what kind of information it seeks—and then decomposes it into targeted sub-questions. Unlike prior methods that operate on a fixed image representation, RVI maintains a Visual Inventory, a dynamically evolving repository of retrieved evidence that is updated with new crops and intermediate answers at each step. A sufficiency query checks whether the accumulated evidence is adequate to answer the original question; if not, reasoning continues with the expanded input. In contrast to static chain-of-thought approaches such as Visual CoT Shao et al. (2023) and ICoT Gao et al. (2025), RVI redefines reasoning as iteratively transforming the input itself, making the process explicitly evidence-driven. We evaluate RVI with motivation-aware decomposition in the open-ended VQA setting on challenging subsets of GQA and A-OKVQA, constructed from instances where state-of-the-art VLMs fail. Importantly, RVI is a training-free approach built on top of these same SOTA VLMs, showing that even strong baselines benefit from our framework. In the open-ended setting, where models must produce exact free-form answers, RVI improves accuracy and, through its transparent steps, makes reasoning failures (e.g. misaligned motivation, reliance on priors, or weak grounding) visible and easier to debug.

We summarize our contributions as follows: (1) We introduce **Reflection with Visual Inventory**, a framework that integrates motivation-aware sub-question decomposition, visual memory, and answer revision; (2) RVI operationalizes focused attention, dual coding, working memory, and motivation awareness in a computational setting; (3) Empirical results across five state-of-the-art VLMs show that RVI improves performance on GQA and A-OKVQA instances where VLMs otherwise fail.

## 2 RELATED WORK

### 2.1 MULTIMODAL CHAIN-OF-THOUGHT REASONING

Recent advances in MLLMs have incorporated Chain-of-Thought (CoT) prompting into vision-language reasoning tasks to improve interpretability and compositional reasoning. Visual CoT Shao et al. (2023) supervises intermediate steps using bounding boxes and natural language descriptions, enabling stepwise textual reasoning aligned with specific image regions. Similarly, CoT-VLA Zhao et al. (2025) extends CoT reasoning into the embodied vision-language-action setting, coordinating perceptual inputs and action planning via CoT supervision. Flamingo Alayrac et al. (2022), while not explicitly CoT-based, exhibits emergent reasoning capabilities in multimodal few-shot settings. Despite these advances, most methods treat reasoning as a linear process and do not include mechanisms for verifying evidence sufficiency or revising incorrect steps during inference.

### 2.2 VISUAL GROUNDING AND HALLUCINATION MITIGATION IN MLLMS

MLLMs are known to hallucinate — producing plausible yet ungrounded outputs — particularly when visual cues are ambiguous or absent Li et al. (2023a). Approaches like LLaVA Liu et al. (2023) and MiniGPT-4 Zhu et al. (2023) partially address this by aligning visual features to language models through instruction tuning and conversation-style data. However, hallucination remains prevalent due to weak or coarse alignment between image regions and answer content. Studies like POPE Otani et al. (2023) and GPT4RoI Chen et al. (2024a) propose region-level probing and grounding modules to expose such failures, but they are primarily diagnostic rather than corrective. Few models incorporate grounding supervision or runtime verification to prevent hallucinated reasoning trajectories.

### 2.3 OBJECT-CENTRIC PERCEPTION IN MULTIMODAL MODELS

Object-centric representations have shown promise in improving the modularity and generalization of vision-language reasoning. Kosmos-2 Huang et al. (2023) introduces grounding tokens to bind textual outputs to detected image regions, while SEED Feng et al. (2023) and GRIT Zhang et al. (2023) leverage dense captioning and segmentation to create object-aware multimodal embeddings. These representations provide finer granularity for interpreting visual input, but are often static and do not evolve dynamically during multi-turn reasoning. Moreover, they seldom support iterative querying or evidence recomposition based on reasoning needs.

### 2.4 STEPWISE AND REFLECTIVE INFERENCE IN MLLMS

Emerging research has begun exploring multi-turn or reflective reasoning in MLLMs. MM-ReAct Yang et al. (2023) integrates tool use with vision-language reasoning, enabling models to retrieve external evidence or re-query APIs. However, reflection is conducted at the macro level, often unrelated to the visual context. Tools-Augmented CoT Gao et al. (2023a) introduces backtracking during language reasoning but does not extend to the visual domain. While recent systems like Prompt Diffusion Gao et al. (2023b) propose image-based reasoning steps, they often operate over the full image and lack localized visual memory or self-correction mechanisms. SCAFFOLD Lei et al. (2025) overlays dot matrices and coordinates on images to better align visual regions with textual references, improving spatial reasoning and reducing hallucination. DDCoT (Duty-Distinct Chain-of-Thought) Zheng et al. (2023) separates recognition from reasoning, prompting LLMs to generate rationales while delegating perceptual grounding to vision models, enabling zero-shot multimodal rationales with stronger generalizability. CCoT (Compositional Chain-of-Thought) Mitra et al. (2024) leverages automatically generated scene graphs as intermediate steps, allowing LMMs to reason over objects, attributes, and relations without requiring explicit scene graph supervision. Our proposed RVI framework departs from static visual conditioning by introducing reflection with a dynamically evolving Visual Inventory.

A closely related line of work is the Interleaved-modal Chain-of-Thought (ICoT) framework, which interleaves visual region features with textual reasoning steps to enrich the multimodal chain-of-thoughtGao et al. (2025). Both ICoT and our proposed RVI share the central insight that reasoning in vision-language models should not be purely textual but must evolve alongside dynamic visual

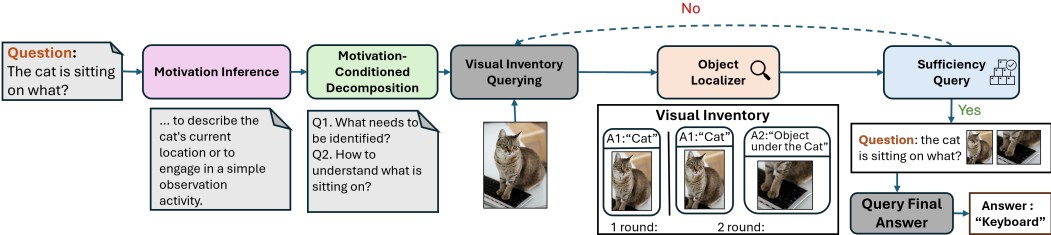

Figure 2: Overview of our proposed framework. Given an input question, the model first infers the underlying motivation, which guides the decomposition of the question into targeted sub-questions. Each sub-question is grounded in the image and answered concisely, with corresponding visual evidence (bounding boxes) stored in a growing visual inventory. This inventory captures the key elements relevant to the main question and provides a transparent record of the reasoning process. A sufficiency query is then used to reflect whether the visual inventory is adequate to answer the main question. If sufficient, the final answer is produced; if not, reasoning continues with the next sub-question. By making motivation explicit, updating evidence step by step, and exposing sufficiency checks, our method turns the reasoning trajectory into a structured, interpretable process for improved performance and systematic debugging of VLM reasoning.

grounding. However, the two approaches differ in emphasis and mechanism. ICoT focuses on integration, progressively inserting attention-driven region features into the reasoning sequence to strengthen multimodal coherence. In contrast, RVI emphasizes verification and debugging: rather than enriching the chain with region-level attention, RVI performs motivation-aware question decomposition, builds an explicit visual inventory of semantically meaningful crops, and employs a sufficiency verifier to determine whether current evidence is adequate before proceeding. This distinction highlights complementary strengths—ICoT enriches reasoning trajectories with finer-grained multimodal context, whereas RVI structures them into interpretable, self-corrective loops designed to reduce hallucination and expose reasoning failures. However, reflective MLLMs with persistent visual memory and fine-grained control over image regions remain an underexplored area.

## 3 METHOD

We propose a motivation-aware visual reasoning framework for VQA that goes beyond chain-of-thought prompting by embedding an explicit mechanism of **reflection**. The guiding steps behind our framework are **Motivation Inference**, ability to correctly infer the intent behind the question, **Contextual Decomposition**, generate sub-questions conditioned on the motivation, ensuring that reasoning focuses on the relevant aspects of the image, **Visual Inventory based Querying**, retrieval of visual evidence and provide intermediate answers grounded in the image, and **Integrative Reasoning with Reflection**, combining grounded sub-answers into a coherent final response while continuously checking if the accumulated evidence is sufficient. An overview of our framework is shown in Figure 2.

### 3.1 MOTIVATION-AWARE DECOMPOSITION

Let $I$ denote an input image and $Q$ a high-level natural language query about the image. The proposed framework introduces an explicit motivation inference stage prior to question decomposition and reasoning.

**Motivation Inference.** We define the motivation $M$ of a query $Q$ as the latent intent or underlying purpose of the question, i.e., the type of information being requested and the rationale for asking. Formally, we compute:

$$M = \text{Motivation}(f, Q) \tag{1}$$

where $\mathcal{Q}$ is the space of natural language questions and $\mathcal{M}$ is the space of motivations. In practice, $f$ is instantiated as a VLM prompted with the query: ``What is the motivation behind

the question?'' This explicit inference of $M$ serves two purposes: (i) it grounds subsequent reasoning steps in the appropriate context, and (ii) it exposes mis-alignments that may otherwise lead to spurious reasoning or hallucination.

**Decomposition.**    Given $(Q, M)$, we again ask the VLM to generate a sequence of at most 5 intermediate sub-questions:

$$q_1, q_2, \ldots, q_T = \text{Decompose}(f, Q, M), \tag{2}$$

where each $q_t$ targets a specific sub-aspect of the original query (e.g., object identification, spatial relation, contextual grounding). The prompt to the VLM is ``From your guess on the motivation, your task is to break down the question into at most [T] sub-questions that step-by-step visually guides attention towards the final answer. Questions should start with "Who", "What", "Where" or "How". Questions MUST NOT start with "Is", "Are" or "Can". Return only the list of questions in JSON format. Question: [Q] Sub_questions: ''

For example, to answer the question "What is located on top of the paper?" we first identify the motivation behind the question, "To identify or describe an object or feature physically placed at the top of a paper — likely testing observation skills, attention to detail, or context awareness." As motivations are made explicit, we can debug decomposition: mismatches between the motivation $M$ and sub-questions $\{q_1, q_2, \ldots, q_T\}$ are easy to spot and fix.

**Visual Inventory based Querying with Reflection**    At each step $t$, the VLM $f$ predicts an intermediate answer $a_t$ and proposes a new crop $i_t \subseteq I$, both conditioned on the current sub-question $q_t$, the full image $I$, prior sub-questions and answers and the latest evidence $V_{1:t-1}$:

$$(a_t, i_t) = \text{Query\_SubQuestion}(f, q_t, a_{1:t-1}, I, V_{1:t-1}). \tag{3}$$

All retrieved regions are stored in a dynamic memory structure we term the Visual Inventory $V_{1:t} = \{i_1, i_2, \ldots, i_t\}$. This progressively constructed repository not only enables iterative attention control and memory-based reasoning but also serves as the substrate for **reflection**, allowing the model to reconsider whether current evidence is sufficient before moving forward. Regardless of whether the final answer is correct, the sequence of $(a_t, i_t)$ across steps provides an interpretable trace of the model's reasoning process, exposing how it grounded sub-questions in visual evidence and revealing both strengths and failure modes in its image–question understanding.

This formulation differs from prior approaches such as VisCoT Shao et al. (2023) and CoT-VLA Zhao et al. (2025), where region-level features are used primarily for supervision or auxiliary explanations, but not as an explicitly constructed sequence of visual queries integrated into a reflective reasoning process.

## 3.2 REFLECTION VIA SUFFICIENCY QUERIES

Our framework incorporates a **reflection phase** after each reasoning step, designed to allow adaptive control over reasoning depth. After producing an intermediate answer $a_t$ for sub-question $q_t$, we ask the VLM $f$ to evaluate whether the accumulated evidence $V_{1:t}$ is already sufficient to answer the original query $Q$.

We define a binary sufficiency variable to store the answer of the sufficiency query:

$$s_t = \text{Query\_Sufficiency}(f, Q, V_{1:t}), \quad s_t \in \{\text{Yes}, \text{No}\}. \tag{4}$$

The sufficiency query is instantiated with the prompt: ``Given the current visual evidence $[V_{1:t}]$, can you answer the original question $[Q]$?'' The sufficiency query act as a reflective debugger, exposing when the visual inventory is inadequate and preventing silent propagation of errors.

If $s_t = \text{Yes}$, the process halts and the final answer is generated as:

$$A = \text{Answer}(f, Q, V_{1:t}, a_{1:t}). \tag{5}$$

Otherwise, if $s_t = \text{No}$, reasoning continues with the next sub-question $q_{t+1}$ as per Equation (3). This reflective mechanism allows the model to adjust its reasoning depth to the difficulty of each query in contrast to prior works Zhou et al. (2023) that advance deterministically through all sub-questions. Finally, the reasoning trace containing the sub-question $q_t$, the intermediate answer $a_t$, the accumulated visual inventory $V_{1:t}$, and the sufficiency verification signal $s_t$ doubles as a debugging log, revealing exactly how evidence, sub-questions, and sufficiency checks contributed-or failed to contribute-to the final answer.

## 4 RESULTS AND DISCUSSION

### 4.1 DATASET CONSTRUCTION

Modern VLMs have demonstrated remarkable progress in image understanding, reasoning, and visual question answering (VQA), in many cases even surpassing human performance on widely used benchmarks. We evaluate our method on two main families of VLM models: Qwen2.5 Team (2024) (Qwen2.5-VL-3B-Instruct, Qwen2.5-VL-7B-Instruct) and InternVL Chen et al. (2024b) (InternVL2.5-8B, InternVL2.5-8B-MPO, InternVL3-8B), chosen due to their strong native grounding capabilities. In terms of dataset, we create challenging subsets for each VLM from two datasets. Specifically, we use the GQA dataset Hudson & Manning (2019), and extract $1,000$ validation examples for each VLM where it fails to predict the answer accurately. This results in a set of failure questions unique to each model. We perform a similar analysis on A-OKVQA Schwenk et al. (2022), extracting an average of 757 examples for each VLM model separately. Some of these selected examples are shown in Figure 3.

We focus on GQA Hudson & Manning (2019) and A-OKVQA Schwenk et al. (2022) because they expose complementary reasoning challenges. GQA emphasizes structured, compositional reasoning and spatial inference, making it well-suited to test whether motivation-aware decomposition yields principled sub-questions. In contrast, A-OKVQA requires integrating visual evidence with external knowledge, surfacing failures that arise when models must reason beyond the image. Together, they provide a broad and challenging testbed for analyzing and debugging VLM reasoning.

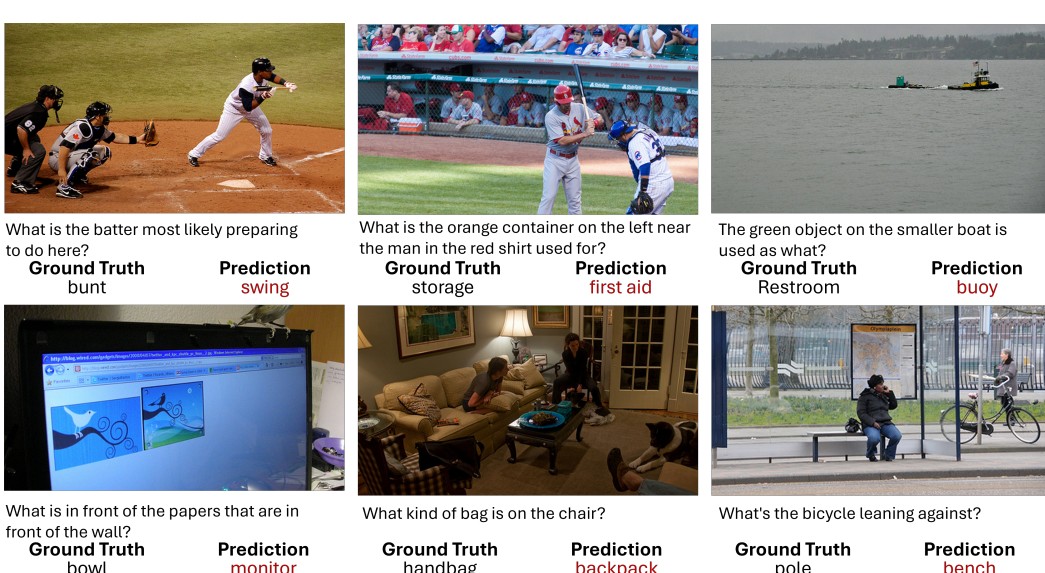

Figure 3: Qualitative samples of failure cases in zero-shot evaluation. Samples are selected from both A-OKVQA (top) and GQA (bottom) evaluated with Qwen2.5-VL-7B-Instruct Team (2024).

**Evaluation metrics.** Instead of adopting multiple-choice style evaluation as with standard VQA, we choose open-ended evaluation, where models must generate free-form answers rather than selecting from predefined options. This evaluation provides a stricter and more realistic assessment of a model's reasoning and grounding ability. This setting is also closer to real-world use, where

answers cannot be reduced to a fixed candidate list. However, evaluating open-ended VQA answers is more challenging because predictions can vary in length, phrasing, or synonyms. To obtain a precise and reproducible measure, we adopt the Exact Match accuracy metric, which considers a prediction correct only if it exactly matches the ground-truth answer string after normalization (e.g., case folding, punctuation removal).

## 4.2 EFFECTIVENESS OF MOTIVATION AWARENESS

We investigated whether incorporating images into the motivation inference process could enhance model performance. Intuitively, providing visual context during motivation inference might help the model better align the inferred intent with the actual content of the image. However, our findings reveal the opposite trend: when images are supplied, the performance degrades (see Figure 4).

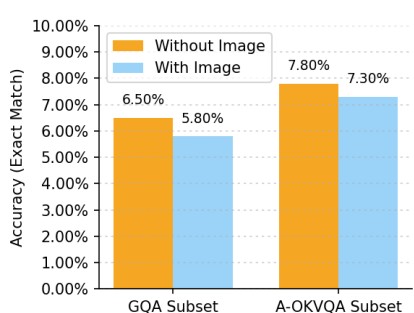

We hypothesize that these results reflect the fact that VLMs are trained to map paired image–question inputs directly to answers but not trained to generate motivation as shown in Figure 5. Therefore, motivation inference should be performed from the question alone, without conditioning on the image, because motivations typically reflect abstract, commonsense intent rather than concrete visual evidence.

Table 1 compares our Motivation-Aware decomposition against sub-questions generated from VisCoT Shao et al. (2023) on GQA. VisCoT provides detailed reasoning steps in the form of instructions to be used in multi-turn QA pipeline. We adapt these reasoning steps by reformulating them as questions leveraging LLM, specifically Qwen2.5-32B-Instruct Team (2024) to create generic sub-questions (e.g., "What objects are in the image?", "Where is the person?"). These sub-questions loosely guide attention, and when combined with RVI this yields clear improvements over the baseline. However, these sub-questions are often shallow and sometimes mis-

Figure 4: Effect of incorporating images during motivation inference on answering accuracy using Qwen2.5-VL-7B-Instruct Team (2024) on both GQA and A-OKVQA subsets.

aligned with the actual intent of the main question. In contrast, our motivation-aware decomposition generates sub-questions that are explicitly conditioned on the inferred motivation leading to more targeted and contextually relevant reasoning steps, resulting in accuracy gains in almost all VLM models as shown in Table 1.

| | Image | | | |
|---|---|---|---|---|
| **Image** | 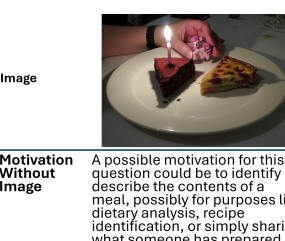 | 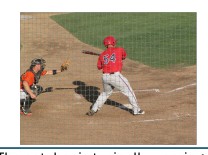 | 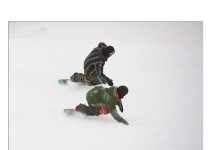 | 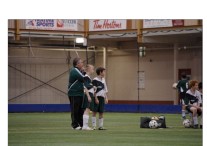 |
| **Motivation Without Image** | A possible motivation for this question could be to identify or describe the contents of a meal, possibly for purposes like dietary analysis, recipe identification, or simply sharing what someone has prepared or eaten. | The catcher is typically wearing protective gear, which includes a helmet, chest protector, shin guards, and sometimes a face mask. A possible motivation for the question could be to understand the equipment used in baseball or softball to ensure the safety of the catcher during play. | A possible motivation of this question could be to identify someone in a group photo, to find out who is participating in an event where caps are worn, or to recognize a person in a particular situation where wearing a cap is significant. | A possible motivation behind this question could be to gather information about the individual, perhaps for a narrative purpose (e.g., a story or a film), to understand their role or significance in the context of the scene, or to infer something about their situation or emotions based on their presence and posture. |
| **Motivation With Image** | The plate contains a slice of chocolate cake with a lit candle and a slice of cheesecake. A possible motivation for the question could be to identify the types of desserts on the plate or to inquire about the occasion, as the lit candle suggests a celebration. | The catcher is wearing a helmet, chest protector, and shin guards. A possible motivation for the question could be to identify the protective gear worn by the catcher during a baseball game, which is essential for safety and protection. | man in front | The person sitting on the bench is a player. A possible motivation of the question could be to identify the player's role in the game or to gather information about their performance. |

Figure 5: Qualitative samples on motivation inference with and without images. Samples are generated using Qwen2.5-VL-7B-Instruct Team (2024)

Table 1: Effectiveness of Motivation awareness in producing relevant sub-questions on the GQA subset. Exact match accuracy (%) is reported. Results are not directly comparable across models, as the subsets consist of failure cases unique to each model.

| Method | Qwen2.5-VL-3B-Instruct | Qwen2.5-VL-7B-Instruct | InternVL2.5-8B | InternVL2.5-8B-MPO | InternVL3-8B |
|---|---|---|---|---|---|
| Image + Question → Answer | 0.0 | 0.0 | 0.0 | 0.0 | 0.0 |
| Multi-turn QA (VisCoT Shao et al. (2023)) + RVI | 6.9 | **6.5** | 10.4 | 12.6 | **12.2** |
| Multi-turn QA (Motivation-Aware) + RVI | **8.7** | **6.5** | **11.7** | **13.1** | 11.8 |

## 4.3 EFFECTIVENESS OF REFLECTION WITH VISUAL INVENTORY

Table 2 evaluates the effectiveness of our proposed Reflection with Visual Inventory (RVI) across multiple VLMs, using exact match accuracy on failure-case subsets drawn from GQA and A-OKVQA. Results are not directly comparable across models, as each subset consists of cases where the respective model failed under zero-shot evaluation.

On the GQA subset, Table 2(a) shows that adding multi-turn QA with motivation-aware (MA) decomposition while using the original image yields modest gains, showing that step-wise reasoning improves over direct answering. Restricting the input further to *cropped image*—equivalent to maintaining a visual inventory of size 1 containing only the crop from the previous sub-question—produces improvements for most models, as it enforces more localized attention. However, the best results are achieved when multi-turn QA is combined with RVI, which systematically builds a visual inventory of evidence across steps and integrates sufficiency queries. RVI consistently outperforms both the original and cropped image settings, indicating that iterative reflection and evidence accumulation provide stronger grounding and more reliable reasoning.

On the A-OKVQA subset questions require external knowledge in addition to visual grounding, incorporating RVI gives consistent improvements across all VLMs as shown in Table 2(b). This demonstrates that our framework is broadly effective at turning failure cases into recoverable reasoning trajectories. The relative gains are smaller than on GQA, likely reflecting the knowledge-intensive nature of A-OKVQA, but the improvements highlight RVI's role in helping models leverage visual grounding more systematically.

Table 2: Effectiveness of RVI across multiple VLMs. Exact match accuracy (%) is reported. Results are not directly comparable across models, as the subsets consist of failure cases unique to each model.

| (a) GQA subset | | | | | |
|---|---|---|---|---|---|
| Method | Qwen2.5-VL-3B-Instruct | Qwen2.5-VL-7B-Instruct | InternVL2.5-8B | InternVL2.5-8B-MPO | InternVL3-8B |
| Image + Question → Answer | 0.0 | 0.0 | 0.0 | 0.0 | 0.0 |
| Multi-Turn QA(MA) + Original Image | 4.6 | 3.6 | 5.4 | 6.7 | 8.8 |
| Multi-turn QA(MA) + Cropped Image | 5.1 | **7.1** | 7.7 | 7.5 | 8.5 |
| Multi-turn QA(MA) + RVI | **8.7** | 6.5 | **11.7** | **13.1** | **11.8** |
| (b) A-OKVQA subset | | | | | |
| Method | Qwen2.5-VL-3B-Instruct | Qwen2.5-VL-7B-Instruct | InternVL2.5-8B | InternVL2.5-8B-MPO | InternVL3-8B |
| Image + Question → Answer | 0.0 | 0.0 | 0.0 | 0.0 | 0.0 |
| Multi-turn QA(MA) + RVI | **6.6** | **7.7** | **7.8** | **8.4** | **11.7** |

## 4.4 DEBUGGING VISUAL REASONING

Figure 6 illustrates how our framework supports visual reasoning debugging through motivation-aware decomposition and the construction of a visual inventory. For each main question, the model first decomposes the query into interpretable sub-questions, each linked to an answer and a grounded image crop. These cropped regions accumulate in the visual inventory, creating a transparent trail of evidence that reflects how the model attends to relevant entities and relations (e.g., man, shopping bag, on top, attached in Figure 6(a), and child, sitting, motorcycle in Figure 6(b)). By making this process explicit, the system exposes where errors arise: the zero-shot baseline mislabels the shopping bag as a backpack or incorrectly answers child instead of man due to language priors, whereas

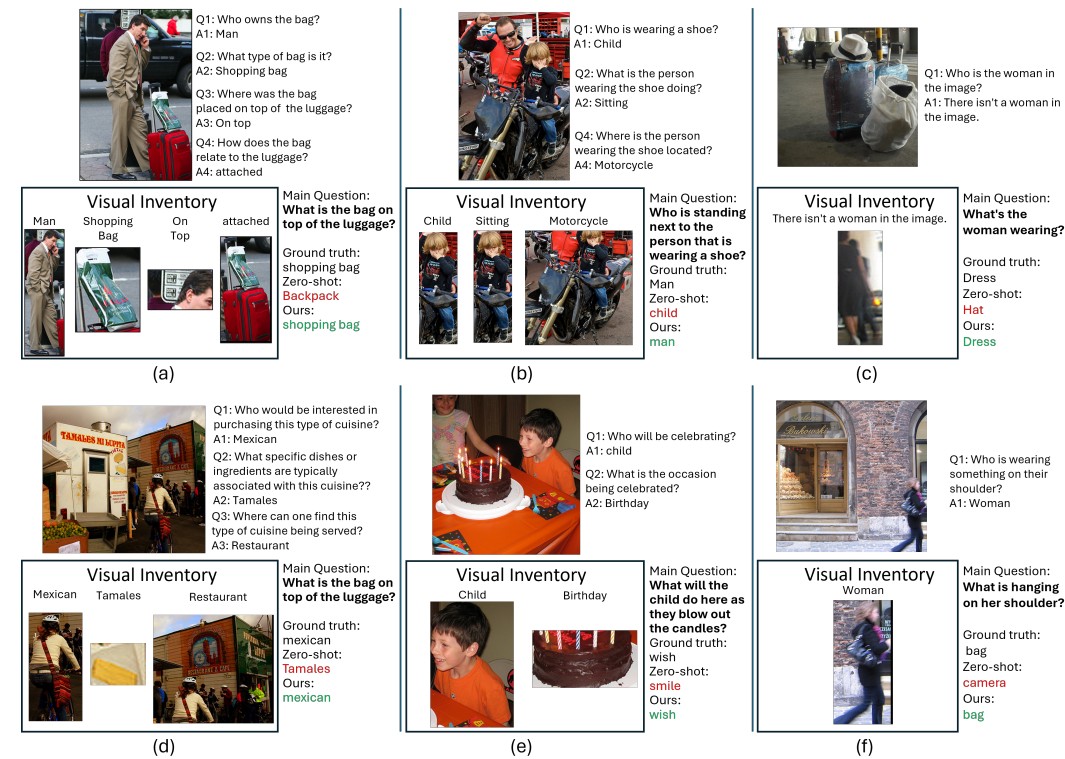

Figure 6: **Visual reasoning debugging with RVI.** The model decomposes questions into motivation-aligned sub-questions with grounded crops, creating a visual inventory that reveals reasoning failures and corrections through transparent evidence trails.

our method grounds reasoning step by step, verifies sufficiency, and ultimately corrects these mistakes. This interpretable reasoning trace not only improves final predictions but also provides an auditable pathway to identify whether failures stem from faulty sub-question answers, missing crops in the inventory, or insufficient verification, thereby turning opaque VQA predictions into a debuggable process. Figure 6(c) demonstrates how RVI prevents language-prior-driven hallucination by introducing a structured, interpretable reasoning loop. Even when intermediate answers are flawed, the iterative visual inventory enables recovery and correction, leading to accurate, evidence-grounded predictions.

## 5 CONCLUSION

In this work, we introduced Reflection with Visual Inventory (RVI), a cognitively inspired framework that integrates motivation-aware decomposition, dynamic evidence accumulation, and sufficiency-based reflection into the visual reasoning process. Unlike prior approaches that treat decomposition as a syntactic exercise or rely on static image grounding, RVI explicitly infers the motivation behind a question, generates targeted sub-questions, and incrementally builds a visual inventory that makes reasoning transparent and debuggable. Across multiple state-of-the-art VLMs, we showed that RVI consistently improves performance on challenging subsets of GQA and A-OKVQA where baseline models fail, demonstrating its effectiveness in mitigating hallucinations and exposing reasoning failures. Beyond accuracy gains, the interpretable reasoning traces produced by RVI provide clear diagnostic signals about whether errors stem from mis-inferred motivation, flawed decomposition, or weak grounding, thus turning opaque model predictions into auditable processes. **Looking ahead, we believe this paradigm of motivation-driven, reflective reasoning with persistent visual memory offers a foundation for building more robust, accountable, and cognitively aligned vision-language system evaluation**.

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
