# OpenReview forum: "Motivation Aware Question Decomposition: An approach to Debugging Reasoning in Vision-Language Models"
_ICLR.cc/2026/Conference — ICLR 2026 Conference Withdrawn Submission_

### Official Review · Reviewer_xoDY · 2025-10-31

**Soundness:** 2
**Presentation:** 3
**Contribution:** 2
**Rating:** 2
**Confidence:** 4

**Summary:**

This paper proposes a question decomposition approach using a multi-turn pipeline to mitigate hallucination issue of VQA.
The approach (RVI) dynamically builds and maintains a Visual Inventor to collect semantically relevant image crops, which are required to verify answers for each step. The Visual Inventor is updated via a reflection phase if required. RVI approach integrates visual grounding directly into the reasoning pipeline, making decomposition and grounding errors visible for systematic debugging. Experiments on GQA and A-OKVQA using Qwen and InternVL models show that RVI improves VQA performance against baselines.

**Strengths:**

- Significance: The paper targets the hallucination of VLM, which is a highly significant issue.

- Clarity: The RVI framework is clearly described, and carefully designed.

- Quality:
  1. The step-by-step process of RVI intrinsically offers better debuggability compared to black-box VLM reasoning, allowing tracing back of errors.
  2. Experiments are conducted using two different model families (Qwen and InternVL) with different scales. This somehow shows the generality of the proposed method.

**Weaknesses:**

1. Combination of existing ideas and lack of novelty: The method can be viewed as an assembly of existing concepts. Question decomposition, CoT reasoning, and reflection mechanism are all well-established. The Visual Inventory is conceptually similar to a dynamic memory or scratchpad, and the sufficiency query resembles a self-correction or search heuristic (e.g., Least-to-Most prompting with a visual check). While the combination is non-trivial, the paper needs a more thorough discussion (and potential ablation) to isolate the original contribution of the proposed RVI compared with the employed components.

2. Insufficient Analysis:
  - The paper explicitly states the motivation on reducing VLM hallucination at the very beginning, yet the core experimental evaluation primarily focuses on VQA accuracy improvement. A critical weakness is the lack of a dedicated, quantitative analysis of hallucination reduction. The authors should have included metrics such as the percentage reduction of non-grounded factual errors (hallucinations) compared to baselines on a subset of known problematic questions, instead of only showing aggregated VQA scores. Or other metrics that can indicate the improvement of hallucinations.
  - Similarly, this paper claims that RVI helps with debugging VLM reasoning, but only providing some cases without quantitative analysis.

3. Missing Evaluation on Proprietary Models:
  - Since RVI is a training-free approach that uses prompting for decomposition and reflection, it is ideally suited for closed-source, state-of-the-art proprietary models (e.g., GPT, Gemini, Claude). The absence of experiments on these models significantly limits the paper's generalizability and its relevance to the current frontier of VLM research.
  - However, it is acceptable if the proposed method only works for smaller opensource models, but achieves comparable final results to proprietary models, as long as sufficient discussion are provided. Or if authors provide persuasive motivation for not discussion proprietary models except for the budget limitations.

4. Scalability and Overhead: The iterative, multi-step nature of RVI inherently incurs high computational overhead. The current presentation lacks an essential quantification of the increase in inference latency compared to the baseline VLM, which is critical for assessing the method's practical utility.

**Questions:**

Please refer to the weaknesses section, and especially response to:
1. The quantitative analysis of debugging and hallucinations. (weaknesses#2)
2. Comparision with proprietary models, or motivations for not using proprietary models. Since this is a training-free approach, it is intuitive and natural to discuss performance on these models. (weaknesses#3)
3. The quantitative results on efficiency. (weaknesses#4)

---

### Official Review · Reviewer_cFFR · 2025-10-31

**Soundness:** 2
**Presentation:** 3
**Contribution:** 1
**Rating:** 2
**Confidence:** 4

**Summary:**

Proposes Reflection with Visual Inventory (RVI), a human-cognition inspired inference framework for VLMs that first performs intent-aware question decomposition, sequentially answers subquestions while building a grounded “visual inventory” over the image, and produces a final answer the model decides it has gathered sufficient evidence. The proposed method is shown to improve performance and interpretability on challenging subsets of the GQA and A-OKVQA datasets.

**Strengths:**

– The paper is well-written and easy to follow

– The human-cognition inspired design is intuitive and clearly motivated

**Weaknesses:**

– The proposed RVI approach is rather complex and computationally expensive, requiring multiple multi-turn inference. While it might be appropriate if used as a “System-2” reasoning framework in conjunction with a fast “System 1” 1-shot method with automated routing, using it unconditionally even for simple straightforward queries seems unrealistic. RVI’s “sufficiency” mechanism could perhaps self-adjust for this, by requiring fewer turns for simpler questions, but the paper does not include experiments showing this.

– There have been numerous prior efforts on using answers to subquestions to improve performance at inference, both for image-to-text and text-to-image generation [A, B, C]. While the paper proposes a more “human cognition” inspired approach, it does not experimentally compare to these works and so it is hard to determine the effectiveness of its proposed changes.

– The proposed “motivation-aware” decomposition strategy appears to generate a list of subquestions and presumably queries the model in order of occurrence, which seems less robust than say a dependency graph representation (like the one proposed in DSG [A]) – Consider the example in Figure 2: If the model is asked “The cat is sitting on what?” on an image not containing a cat at all, it’s not clear that the RVI approach would produce a correct response efficiently (eg. There is no cat at all in the image). In general, it is unclear how RVI would deal with false premise questions, which is a fairly realistic and common real-world scenario.

[A] Cho, Jaemin, et al. "Davidsonian scene graph: Improving reliability in fine-grained evaluation for text-to-image generation.", ICLR 2024
[B] Singh, Jaskirat, and Liang Zheng. "Divide, evaluate, and refine: Evaluating and improving text-to-image alignment with iterative vqa feedback.", NeurIPS 2023
[C] Nguyen et al., Coarse-to-Fine Reasoning for Visual Question Answering, CVPRW 2022

**Questions:**

Please address the weaknesses above, especially around experimental comparison to prior work, the empirical efficiency of the self-determined sufficiency mechanism, and the behavior of RVI on false premise questions.

---

### Official Review · Reviewer_U4z7 · 2025-11-01

**Soundness:** 2
**Presentation:** 3
**Contribution:** 2
**Rating:** 2
**Confidence:** 4

**Summary:**

This paper proposes RVI (Reflection with Visual Inventory), which reduces VLM hallucination through motivation-aware question decomposition and iterative evidence accumulation via image crops. While the motivation-aware decomposition idea is interesting, the paper has critical flaws: evaluation only on pre-selected failure cases (making baselines artificially 0%), modest absolute gains (6-13%), lack of efficiency analysis, and limited technical depth beyond prompt engineering.

**Strengths:**

- The idea is intuitive and well-motivated
- Evaluated on 5 state-of-the-art VLMs

**Weaknesses:**

- Only tests on pre-selected failures (baseline is 0% by construction). No full dataset results, so can't assess real-world value. 6-13% gains mean 87-94% of hard cases still fail
- Requires ~10+ VLM calls per question. Zero discussion of computational cost or inference time
- Only 2 datasets, 1 baseline comparison (VisCoT). Where are comparisons with Tree of Thoughts, self-consistency, other hallucination methods?

**Questions:**

- What is the performance of the proposed model on full dataset?
- Please deliver the detailed discussion of computational cost or inference time
- Why does image-free motivation work better? Current explanation is unconvincing
- Where are comparisons with Tree of Thoughts, self-consistency, other hallucination methods?

---

### Official Review · Reviewer_5EpZ · 2025-11-04

**Soundness:** 3
**Presentation:** 3
**Contribution:** 2
**Rating:** 4
**Confidence:** 3

**Summary:**

The paper proposes Reflection with Visual Inventory (RVI), a framework to reduce hallucinations in Vision-Language Models during VQA. It first infers a question’s underlying motivation, then decomposes it into targeted sub-questions. At each step, the model gathers localized image crops into a dynamic “Visual Inventory”, answers sub-questions, and runs a binary sufficiency check; if evidence is insufficient, it continues iterating. This makes reasoning transparent and debuggable (exposing poor decomposition or weak grounding) and guides attention beyond static, post-hoc methods. Tested with several VLMs (Qwen2.5, InternVL), RVI improves exact-match accuracy on GQA and A-OKVQA, demonstrating gains without training via motivation-aware decomposition, iterative grounding, and reflection.

**Strengths:**

**Originality:** Motivation-aware decomposition and iterative selection of image crops for grounding analysis is moderately novel compared to techniques such as ICoT, DDCoT, CCoT, etc. Direct comparisons against competing methods would strengthen novelty if the method is shown to be an improvement.

**Quality:** Comparisons to strong benchmarks on the two datasets studied are lacking.

**Clarity:** The writing and presentation are clear. However, additional data and experimentation would be welcome.

**Significance:** The technique achieves ~10% on GQA and ~8% on A-OKVQA, evaluated on “a set of failure questions unique to each model”. This is a modest performance improvement on the failure set, and it is unclear how well the method performs in overall accuracy as compared to competing methods. Without clear comparisons to competing methods, it is difficult to evaluate the extent to which the method achieves impressive performance.

**Weaknesses:**

The main weakness is that the technique is not compared to strong benchmarks which decompose question-image inputs in different competing ways. The comparisons are made to weaker versions of the same approach, not to competing similar approaches - which would be a stronger comparison.

Further notes on:
- **Soundness:** Reasonably good experimental setup, although more thorough comparisons to other methods are called for.
- **Presentation:** Presentation is overall good, although some aspects can be made more informative. For example, figure 6 does not make clear what debugging inferences are made from the presented outputs.
- **Contribution:** The idea is solid. However, it is not particularly unique relative to prior work as several other techniques have taken different approaches to select relevant image crops for grounding judgements.

**Questions:**

- Does the method improve over strong comparisons - competing methods rather than weaker versions of the proposed method?

- The authors claim that the method supports debugging visual reasoning. I would want to see worked examples in which it is shown how the trace of intermediate crops is used to diagnose deficiencies in reasoning. Section 4.4 figure 6 are suggestive, but don’t explicitly demonstrate the use of the method for debugging.

---

### Note · Authors · 2025-11-17

I have read and agree with the venue's withdrawal policy on behalf of myself and my co-authors.